# *"When a man drinks alcohol it's cool but when a woman drinks she is a hoe"*: A qualitative exploration of alcohol, gender, stigma, and sexual assault in Moshi, Tanzania

**Alena Pauley**[1], **Madeline Metcalf**[1], **Mia Buono**[1], **Sharla Rent**[1,2], **Mariana Mikindo**[3], **Yvonne Sawe**[3], **Joseph Kilasara**[3,4], **Judith Boshe**[3,5], **Catherine A. Staton**[1,6]*, **Blandina T. Mmbaga**[1,3,5]*

1 Global Emergency Medicine Innovation and Implementation Research Center, Duke Global Health Institute, Duke University, Durham, North Carolina, United States of America, 2 Duke Department of Pediatrics, Duke University Medical Center, Durham, North Carolina, United States of America, 3 Kilimanjaro Christian Medical Center, Moshi, Tanzania, 4 Department of Clinical Nursing, Kilimanjaro Christian Medical University College, Moshi, Tanzania, 5 Kilimanjaro Clinical Research Institute, Moshi, Tanzania, 6 Duke Department of Emergency Medicine, Duke University Medical Center, Durham, North Carolina, United States of America

* catherine.staton@duke.edu (CAS); blaymt@gmail.com (BTM)

**Data Availability Statement:** Data are only available upon reasonable request, as participants

## Abstract

Alcohol's ever-increasing global use poses a distinct threat to human well-being, with intake and associated burdens rising especially quickly in low- and middle-income countries like Tanzania. Prior research has shown alcohol use and related consequences differ by gender in Moshi, Tanzania, with important implications for both clinical care and future alcohol-reduction interventions. This study builds upon this knowledge by providing a deeper understanding of how gender differences affect alcohol-related stigma and sexual assault among Emergency Department (ED) and Reproductive Health Center (RHC) patients at Kilimanjaro Christian Medical Center (KCMC) in Moshi. In-depth interviews were conducted among ED and RHC KCMC patients (n = 19) selected for participation via purposive sampling. A mix of inductive and deductive coding schemes was used to identify themes and subthemes. All data were analyzed through a grounded theory approach. Gender roles that linked men with financial responsibilities and women with child caretaking led to different expectations on alcohol intake, with alcohol use encouraged for men but vilified for women. Women who drank, for example, were deemed poor mothers and undesirable spouses. Patients likewise emphasized that both alcohol-related stigma and sexual violence disproportionately impacted women, the latter fueled through alcohol use, with serious and lasting acts of discrimination and isolation from community members seen among women alcohol users but not for men. Women alcohol users in Moshi are subject to severe social consequences, facing disproportionate stigma and sexual violence as compared to men. Alcohol-related treatment for women should be mindful of the disproportionate burdens present in this context while treatment for men should be cognizant of the social pressures to drink. Strategies to address and/or mitigate these factors should be incorporated in subsequent care and interventions.

did not consent to public data publishing, and data transfer requires a written agreement approved by Kilimanjaro Christian Medical Centre Ethics Committee and the National Institute for Medical Research (Tanzania). Data inquiries can be sent to Gwamaka W. Nselela at gwamakawilliam14@gmail.com.

**Funding:** This project was funded by the Duke Global Health Institute Graduate Student funds (AMP), and the Josiah Trent Foundation (21-06 to CAS). These two financial awards funded the salaries of JK, YS, and MMi as research assistants hired specifically for this study. No other authors received specific funding for this work. Infrastructure built by an NIH grant (R01 AA027512 to CAS) was used to support the data collection and analysis processed for this grant to understand gender-related aspects of alcohol use at KCMC. The funders had no role in study design, data collection and analysis, decision to publish, or preparation of the manuscript.

**Competing interests:** The authors have declared that no competing interests exist.

## Introduction

Alcohol is a significant and growing threat to human health, implicated in 1 in 20 deaths worldwide and accounting for 5.1% of the global burden of disease and injury annually [1]. In recent years, low- and middle-income countries (LMICs) have been facing exceptionally rapid increases in consumption and ensuing harm, with alcohol now established as the leading cause of disease burden in many Sub-Saharan African countries [2, 3]. Alcohol use behaviors in Africa are also among the most hazardous worldwide, with this region having especially high rates of binge drinking and alcohol dependence that produce greater health risks than moderated intake [2, 4, 5]. While already concerning, harm is concentrated more still in countries like Tanzania, which have almost twice the incidence of heavy episodic drinking and alcohol use disorder than neighboring Kenya or Mozambique [1]. In Northern Tanzania, excessive intake is fueled in part by social customs that promote alcohol use from a young age, combined with alcohol's low cost, rising levels of disposable income, and ready availability [6].

Gender is an essential modifier in examining risk factors of disease and injury, clinical progression, and responses to treatment across a wide range of health conditions, including alcohol abuse [7, 8]. As follows, treatment options should integrate biological and sociocultural gender nuances in order to effectively provide care [9–12]. The need to consider gender in health is heightened in many LMICs, given women's more vulnerable social status [13–15]. In these contexts, women often are relegated to unpaid household and child caretaking responsibilities and have less decision-making power, education, employment opportunities, and access to healthcare services than their male counterparts [16–18]. These stringent socially enacted distinctions between men and women in some LMICs make it even more important to consider gender differences in creating more effective alcohol-reduction programs and clinical treatments.

Looking at gender and alcohol use worldwide, men are known to consume more alcohol and account for three times more alcohol-related harm to themselves and others [19–22]. Men are also more likely to be diagnosed with alcohol use disorder (AUD) in their lifetime [23], hospitalized for alcohol poisoning [24], die in traffic crashes involving alcohol [25], and die from other alcohol-related causes [21]. On the other hand, women are disproportionately burdened with chronic health harms, including more rapid alcohol-related cognitive decline and brain atrophy [9], and greater susceptibility to alcohol-induced liver disease [26], cardiovascular disease [9], and certain cancers [27–29]. In many settings, women also face greater social consequences related to alcohol use, including heightened stigma [30], discrimination, and sexual violence [31].

The stigma around substance use disorders is an especially notable gendered consequence as it stands as a barrier to treatment delivery [32, 33]. Past research based in Moshi, an urban town in Northern Tanzania, has found that gendered differences underlying alcohol-related stigma exist and disproportionately burden women [30]. This is concerning given that stigma is an established barrier to treatment, thus potentially hindering women in this region, especially, from receiving needed care [30, 34]. The stigma surrounding women's alcohol use in Tanzania has also been interlinked here with sexual assault; for example, a Dar es Salaam-based study has found that women who drink alcohol are perceived to lack morals and are considered prostitutes regardless of their occupation [35]. Alcohol abuse among women in Moshi has also been associated with a history of sexual violence and a greater incidence of testing positive for sexually transmitted diseases (STDs) [36].

Beyond these effects, alcohol use in Moshi is high, with some sources estimating intake to be 2.5 times higher here as compared to nearby regions [37–39]. This high use combined with the lack of resources and trained professionals dedicated to alcohol-related care [40, 41] point

to an urgent need for effective alcohol-reduction interventions in this area. To make these programs effective for both men and women, gender differences in alcohol use and alcohol-related harms should be considered. Literature discussing how these harms may differ by gender, particularly in LMICs, is sparse. Further, the relationship between gender, stigma, and sexual assault is nuanced and complex, a qualitative exploration of these themes is needed to better understand the specific impacts of alcohol use. This paper aims to fill these gaps by first examining gendered expectations of alcohol use and then exploring men's and women's differentiated experiences with alcohol-related stigma and sexual violence in Moshi, Tanzania.

## Methods

### Study design

Beginning in October 2021 and ending in May 2022, data focused on understanding gender differences in alcohol consumption was collected from Emergency Department (ED) and Reproductive Health Center (RHC) patients presenting for care at the Kilimanjaro Christian Medical Center (KCMC) in Moshi, Tanzania. This manuscript stems from a larger mixed-methods study of alcohol use in Moshi, Tanzania [42]. In this analysis, we use qualitative data gathered from in-depth interviews to focus on gender, stigma, and sexual assault. In total, 19 in-depth interviews were collected through purposeful sampling from ED and RHC patients who previously completed surveys.

### Setting

Moshi is an urban town set in Northern Tanzania, bordering Mount Kilimanjaro National Park and Kenya. KCMC, located in Moshi, functions as the referral hospital to over 11 million people [43, 44]. KCMC's ED and RHC were chosen as the two clinical sites for participant sampling to better elucidate 1) risky alcohol use behaviors and 2) gender differences in alcohol use.

Alcohol use is a well-defined risk factor for injuries, especially injuries prompting ED visits [45, 46]. A previous study by our group established that this holds true within Moshi: 30% of injury patients tested positive for alcohol use upon presentation to KCMC's ED for care [47]. Given that KCMC's ED cares for a patient population with known risky alcohol use behaviors, it served as an ideal location to better study alcohol-related harms, and so was chosen as one of the main clinical sites for this study. The RHC at KCMC is the referral unit for all obstetric and gynecological issues in Moshi and its surrounding areas. The RHC's patient population, which consists entirely of females, allowed for data collection to occur in a safe, gynocentric environment to facilitate the open and honest sharing of women's thoughts and experiences with alcohol use. An incidental advantage in using this location as a study site is it allowed us to incorporate women with less extreme alcohol use behaviors, as women seeking prenatal or antenatal care typically had reduced alcohol intake.

### Research team and reflexivity

The research team based in Tanzania consisted of two female (YS and MM) and one male (JK) research assistant. This team was overseen on the ground by one physician-researcher (BM) with extensive previous experience in alcohol-related and epidemiology-focused research. All three research assistants were born and raised in Tanzania, fluent in both Kiswahili and English, had obtained a college degree or higher, had previous experience working in research, formal training in qualitative research practices, and held the strong interpersonal skills needed for conducting an in-depth interview that addressed potentially sensitive topics such as

stigma or sexual abuse. The Tanzanian team was supported by partners in the US with expertise relevant to emergency medicine, alcohol use disorders, and qualitative data analysis.

Of note, while this study was based in Moshi, Tanzania, the first author is not a local researcher. The local perspective has regardless been maintained in this paper as the Tanzanian research team informed all data collection tools and analysis procedures with the relevant sociocultural nuances of the study setting. The decision over authorship placement was made jointly between the Tanzanian and US research teams given that the first author spearheaded all study efforts starting from initial study development through data analysis and the subsequent formation of this manuscript.

## Participant selection and sample size

Enrolled individuals met the following criteria: 1) received initial care at KCMC's RHC or ED, 2) were 18 years of age or older, 3) were conversant in Kiswahili, 4) had the capacity to give informed consent, and 5) were not prisoners. A purposive sampling strategy was employed to select an anticipated 20 interview candidates, or until saturation was reached, from the 678 patients who had first completed quantitative surveys within the larger study. Twenty in-depth interviews were initially anticipated, given that saturation is typically reached at 20 interviews in studies with narrow aims [48]. For this project specifically, five ED women patients, five RHC women patients, and ten ED men patients were targeted to achieve a gender-balanced perspective in collected data, and saturation was defined as a dearth of new information or themes after three consecutive interviews. Saturation was reached for the ED men category at 9 interviews, after which data collection ceased, so in total, 19 in-depth interviews were completed.

Patients were purposefully selected for participation in in-depth interviews to represent varied 1) sociodemographic statuses including age, income, profession, marital status, and religion, 2) levels of alcohol intake, and 3) perspectives on and experiences with alcohol, all of which initially noted during survey collection. The study lead assessed interviewee characteristics en masse monthly during the data collection timeline to ensure the sample remained representative of these targeted attributes. Any needed modifications were implemented immediately following these reviews.

## Participant recruitment and interview procedure

Patients were approached for potential study participation only once and in a quiet, private location of KCMC's ED or RHC after they had been medically stabilized. All those approached were provided the option to decline study participation, but if interested, informed consent was obtained through an in-depth discussion of study goals, procedures, risks, and benefits. If willing to participate, written consent was obtained unless the patient was illiterate, in which case, depending on their ability level, the individual marked their initials or a cross mark.

As alcohol use in Moshi is known to be stigmatized, one-on-one semi-structured in-depth interviews were chosen as the method for qualitative data collection to encourage participants, especially women, to be able to speak freely about alcohol use while also maintaining their privacy. Interview participation was introduced to eligible patients either at the conclusion of their quantitative survey or in a later phone call. If a patient was interested and willing to participate, a later appointment to meet was scheduled at the patient's convenience. All interviews were completed in quiet rooms at KCMC with only the patient and the research assistant present, and with both members of this pair identifying as the same gender. This gender-matching was done to foster patients' truthful reporting of their opinions on alcohol use. Interviews were all conducted in Kiswahili, generally lasted from an hour to an hour and a half and were

digitally audio-recorded. Directly after the initial collection, audio recordings were transcribed into written Kiswahili and then translated into English by a member of the Tanzanian research team. Participants were given a small stipend to reimburse travel costs and offered snacks midway through their interview.

In-depth interviews were semi-structured, using an interview guide that had been organized across the following six domains: 1) community perspectives and expectations of alcohol use, 2) men's alcohol use, 3) women's alcohol use, 4) differences in alcohol use between men and women, 5) alcohol use during pregnancy, and 6) participant's recommendations for future alcohol-reduction programs and interventions. If content arose during the interview that was either confusing, contradictory to a previous statement, or especially notable, the interviewer asked probing follow-up questions to better elucidate the patient's meaning or expand upon a unique experience with alcohol. The interview guide had been first developed in English using a team-based approach in collaboration with both the US and Tanzanian research teams, was translated into Kiswahili, reviewed for cultural appropriateness, pilot tested, and revised into its final form.

## Data analysis

A grounded theory approach was employed in this analysis to allow new themes and ideas to arise organically, given that this topic has been sparsely studied in this particular setting [49]. A first draft of our codebook was deductively and inductively developed once the first four interviews had been completed. The interview guide was used to deductively outline broad categorical domains as described previously. Following this and further interview transcript analysis, the research team inductively generated specific themes from transcripts without preconceived theoretical hypotheses in concordance with grounded theory methodology [50]. These themes were amassed into a codebook which was extensively discussed and reviewed between the main analyst, AP, and the Tanzanian research team, YS, MiM, and JK. Subsequent changes were made to maintain cultural relevance and accuracy to the codebook while also ensuring content validity. The codebook remained an evolving, non-linear document that was updated as novel themes surfaced throughout data collection [51, 52].

AP provided training to YS, MiM, and JK on interview coding. All members of this group independently coded the initial interviews, meeting after each interview was coded to compare coding strategies and maintain coding consistency. Any differences that arose were discussed among this group until a unanimous agreement was reached. The coding team replicated this process for each consecutive interview until 80% agreement was obtained. After this point, the main analyst then used the codebook to complete the coding of all remaining interviews, a decision approved by the research team. Memos, developed based on codebook themes, served as the foundation for the findings section below. As is characteristic of grounded theory research, data analysis and participant sampling occurred in parallel so that patients holding specific characteristics of interest could be preferentially selected [50, 51]. For example, as part of the larger study, the effect of alcohol use on mental health was an important theme noted early on, so the research team purposefully enrolled patients screening positively for both depression and alcohol use disorder to participate in subsequent interviews.

Given the many existing definitions and types of stigma in relation to mental health and substance abuse, in this study, we defined stigma as the negative and often unfair beliefs surrounding a specific population or attributable to a certain characteristic – here, alcohol users and alcohol use. Perceived and experienced stigma were the two main types that emerged in in-depth interviews. *Perceived stigma* is one's beliefs on how the community views a certain attribute [53], surfacing in this study when participants described alcohol-related stigma's

presence in their community and compared its power between genders. In contrast, *experienced stigma* refers to concrete encounters with any stigmatizing behaviors and beliefs [54]. This arose in patients' re-telling of friends, relatives, or acquaintances' lived experiences with alcohol-related stigma. As noted in our other publication, while this analysis is centered on examining gender differences, participants self-identified by biological sex [42]. Tanzania is reported to hold little gender diversity, so patients identifying as female were categorized as women, and those identifying as male were categorized as men.

## Ethics statement

The Tanzanian National Institute of Medical Research, Duke University Institutional Review Board, and Kilimanjaro Christian Medical University College Ethical Review Board all granted ethical approval for this study prior to the start of data collection. Data was de-identified as much as possible and shared through a data share agreement. While personal health information was utilized in screening and enrollment processes, data was de-identified when collected, stored, and analyzed.

## Findings

Nineteen in-depth interviews were collected from participants whose demographics spanned a wide range of age, income, and education statuses, along with differing levels of alcohol intake as previously documented [42]. The interplay of gender on alcohol-related stigma and sexual violence was explored with a focus on elucidating the differences between men's and women's experiences. The analysis yielded four main themes: Gender Roles, Alcohol's Impact on Gendered Responsibilities and Family Life, Alcohol-Related Stigma, and Alcohol and Sexual Violence (Table 1). In the first theme, men were perceived to play a financial support role, while women held primary childcare responsibilities in a traditional household. These roles led to unique social expectations for alcohol intake and fostered more negative connotations around women who drank, such as the perception that they would be poor mothers and undesirable to marry. Alcohol-related stigma and sexual assault were likewise found to disproportionately burden women, leading to social isolation and a perception that women who drink are able

**Table 1. Themes and subthemes of patient interviews.**

| Themes | Sub-themes |
|---|---|
| Gender Roles | Traditional responsibilities for men and women |
|  | Dissimilar expectations for alcohol intake |
| Alcohol's Impact on Gendered Responsibilities and Family Life | Women who drink are perceived as poor mothers |
|  | Men who drink but can still provide financially face less social censure |
|  | Women who drink are undesirable to marry |
| Alcohol-related Stigma | Women face more severe alcohol-related stigma than men |
|  | Lack of discrimination and isolation surround men who drink |
|  | Discrimination and isolation surrounding women who drink |
|  | Community support and implications on women's health |
|  | Stigma's influence on sexual violence |
| Alcohol and Sexual Violence | Women face disproportionate alcohol-related sexual harm |
|  | Alcohol is used as a tool to advance sexual acts |
|  | Women's alcohol use is associated with sex work |

and prone to be sexually assaulted. Finally, alcohol was noted as an accelerant through which men could facilitate sexual advances with women, using alcohol to remove women's inhibitions, limit their decision-making capacities, and encourage risky behaviors.

## Gender roles

While never formally asked during interviews, sociocultural gender roles were nevertheless a prevailing theme noted to underlie the gendered consequences of alcohol use. While these norms did not apply equally to all individuals in Moshi, the presence of this theme in most interviews, especially unsolicited, suggests an enduring social significance. As explored below, men's and women's different responsibilities within a traditional Moshi household fostered dissimilar community expectations for their behavior and alcohol intake.

**Traditional responsibilities for men and women.**   Men were described as having a "*responsibility to look after the family by seeking money*" (Interview #13, Female), and were seen to shoulder most of the fiscal responsibilities for families in Moshi. This was achieved through outside employment – physically removing them from the home – to ensure the financial "*protection*" and "*provision*" of their households – "*for the most part family responsibilities especially involving money are of a man*" (Interview #8, Male).

In contrast, respondents expressed that women's roles lay in safeguarding home and family life, especially as it relates to caring for their children. Women, in general, were seen as "*responsible for rearing children*" (Interview #12, Male), "*pillars in the family*" (Interview #3, Female), and "*under men's authority*" (Interview #4, Male). Others described women as the "*passer of family norms and love*"; "*in most families, women are the one taking care of children*" (Interview #8, Male). In opposition to men, this child-caretaking role physically tied women to their homes but also provided them with little free time as this job requires near constant time and attention.

Building on this sentiment, one participant mentioned that higher, potentially unrealistic, expectations exist for women:

> "*Our African society has a picture of a very perfect woman without any kind of flaw. They think that a woman should be perfect at everything in order to be a woman. . . forgetting that women are human beings too and no human is perfect. Having flaws doesn't make one less of a woman, but unfortunately, our society doesn't understand that*" (Interview #10, Female).

**Dissimilar expectations for alcohol intake.**   These gender roles directed many participants' belief that men and women should drink different amounts of alcohol. In particular, it was thought that women should drink less because alcohol consumption prevented women's childcare responsibilities from being fulfilled – "*the effects tend to be [more] severe for women than to men because women have more sensitive roles in community development, specifically in raising children and family*" (Interview #7, Female). As summarized by another: "*the community doesn't expect [women] to drink alcohol as they have a lot of activities and home chores. . .and look after kids*" (Interview #4, Male).

In contrast, men's alcohol use was generally tolerated by women and community members.

> "*Society's expectations and duties differ between men and women, making the effects of alcohol more severe and sensitive to women because society depends on them for the high task of family nurturing. For men, after when they have secured money for the family, things tend to run smoothly even when they are drunkards, unlike women who fail to take care of the family*" (Interview #7, Female).

These gender roles and dissimilar expectations of alcohol use may stem from Moshi being, as one participant described, *"a patriarchal society who sees it as okay for a man to drink alcohol. . .But they treat women who drink alcohol very different from how they consider themselves"* (Interview #9, Female).

## Alcohol's impact on gendered responsibilities and family life

Respondents noted that contrasting gender responsibilities and alcohol use expectations created unique consequences when a mother versus a father drank in excess. One respondent summarized the impact on families:

> *"If the father is a drunkard, the family will find it very difficult to meet their [financial] needs. But the responsibility of the day-to-day upbringing of children in the family is of a woman. If the mother is drunkard, the children will miss out on essential services such as hygiene and cooking healthy food"* (Interview #8, Male).

As a result of these unique impacts on families, social perceptions of mothers and fathers who drank were likewise dissimilar.

**Women who drink are perceived as poor mothers.** A recurring question arose as a byproduct of traditional gender roles: *"Women are the caretakers of the family; if they drink too much alcohol, who will look after the family?"* (Interview #13, Female). Answering this, a common perception of women who drank alcohol was that their behavior would negatively interfere with childcare responsibilities – *when a woman embraces too much alcohol she cannot take proper care of her family. . .she will no longer be effective because of too much alcohol"* (Interview #3, Female)*; "when a mother is drunk, she can't prepare family food, which have an impact on the mental health of their children"* (Interview #7, Female). As a result, one interviewee noted that *"the women who drink lose their respect and dignity as mothers in the society"* (Interview #16, Female).

A mother's alcohol use was seen to have a stronger impact on family functioning than the fathers because of the importance placed on childcare. As such, strong opinions and greater social censure surrounded women who drank –

> *"Women [who drink] are viewed as morally wrong and bad-mannered. Because the community believes in them and has a lot of expectations, we normally say if the woman is not strong, the whole family is weak. So once a woman drinks, they will be isolated and discriminated against by the members of the community"* (Interview #19, Female).

**Men who drink but still can provide financially face less social censure.** In opposition to social norms around women, a man who drank but could still provide financially for their family was seen to have sufficiently fulfilled his role and was given the proverbial green light to consume alcohol. As one respondent explains, *"when a man has earned money and brings it home, even if he drinks too much alcohol the community finds it okay because he has already fulfilled his responsibility"* (Interview #6, Female). One participant even described men who go to work intoxicated but face little stigma or social censure due to their pivotal roles as doctors or teachers in the community –

> *"In villages, it's difficult to stigmatize these people because they are one of the few professionals that the whole village depends upon. So even those who drink know that they are important and instrumental in the community and that people depend on them for different services like health or education"* (Interview #3, Female).

**Women who drink are undesirable to marry.**   Interestingly, the association of alcohol use and poor childcare negatively impacted women who drank even before they were married or pregnant. Because of the aforementioned perceptions and impacts of women's drinking, respondents commented that women who consumed alcohol were undesirable for marriage:

*"A woman's drinking habit may cause a delay or difficulty in getting married because men are watchful of the characters of women they want to marry. If they see that a particular woman is into alcohol, they [will question] if she will be able to perform duties and activities as a married woman, as a mother to her future family. . .Men are scared of being left alone with family and children when the woman is busy out there drinking alcohol"* (Interview #10, Female).

The undesirability of women who consume alcohol also relates to community assumptions of sexual promiscuity, which further makes women eligible for a good marriage. This was especially true if women drank hard liquor, rather than beer. As one participant explained:

*"When a man drinks alcohol it's cool but when a woman drinks alcohol [she] is a hoe. A man who works, drinks alcohol, and can provide, he is husband material. But a woman who works, drinks alcohol, and can provide, she is not wife material. . .Myself at the university, someone can tell me a joke like 'who will marry a woman like you who drinks hard liquor,' well that is a stigma because if I drink hard liquor no one will marry me."* (Interview #15, Female).

## Alcohol-related stigma

**Women face more severe alcohol-related stigma than men.**   As was alluded to in the quote above, a persistent thread across many interviews was the existence of perceived and experienced alcohol-related stigma. Importantly, in most instances, the stigmatizing, negative opinions around women who drank were more severe than those surrounding their male counterparts – *"women who drink are stigmatized more than men"* (Interview #7, Female). This was true even when accounting for similar or even lesser quantities of alcohol consumed – *"men use alcohol much more than women, but women face much more stigma about alcohol use than men in the community"* (Interview #3, Female). Two respondents described this discrepancy:

*"If I sat somewhere with a man who is the same age, and I, as a woman, drank two beers and the man drank ten beers, the community will find its wrong for me to drink two beers but the man is not guilty for drinking ten beers. So, what the society considers to be inappropriate alcohol use is mostly based on gender, and it is wrong for a woman to drink alcohol even if she is an adult"* (Interview #15, Female).

*"A woman who drinks is viewed as a person that is not self-aware or respectful at all, but a man who drinks is viewed as a successful, independent man because he can afford providing for his family and can afford alcohol too. He is seen as an updated man who has connections with potential people he meets when he goes drinking"* (Interview #10, Female).

While not a unanimous belief – two participants stated that *"stigma exists for both who drinks alcohol, whether a men or women"* (Interview #11, Male) – in general, the words participants used to describe women who drink were harsher and more damaging – *"That value of a respectable woman. . .is no longer there when she drinks too much"* (Interview #7, Female) than those used to describe men.

**Lack of discrimination and isolation surrounding men who drink.** For men, stigma was either less severe "*society excludes them in making important decisions. . .society does not trust them with the work*" (Interview #3, Female) or nonexistent— "*there is no stigma among men who drink alcohol*" (Interview #4, Male). One important excerpt helps illustrate the relationship surrounding men who drink in Moshi and stigma.

> "*There is little stigma towards men who drink alcohol . . . His fellow men will not exclude him, they will collaborate with him, they will laugh and live normally with him. Maybe if his addiction to alcohol. . .damages the job then that is where there will be a problem. . . . When I think of stigmatization, it involves alienating someone and refusing to engage with someone just because of excessive alcohol consumption. This is rare among men in Tanzania. . .So, if you see someone being stigmatized, maybe it's because he drinks too much alcohol and he doesn't have money, but not because of drinking too much alcohol alone*" (Interview #9, Female).

This quote highlights the normalization of excessive alcohol use among men leading to decreased social stigma around their drinking. For men, "*if you are taking care of everything at your place and work, it's never perceived as stigma no matter how much you drink*" (Interview #8, Male).

The perceived stigma that arose from men's alcohol use instead was seen as a poor reflection on their family instead of themselves—"*men who drink too much alcohol are the ones stigmatizing their families for not caring for their families. They spend all they get on alcohol regardless of their families' needs*" (Interview #1, Female). She continued in summarizing this, saying:

> "*Society tends to stigmatize [men] because they have despised their families and gave alcohol the first priority. . .They are seen as silly drunkard people who cannot make decisions. And even children tend to disrespect them saying like someone's father is a drunkard person*" (Interview #1, Female).

**Discrimination and isolation surrounding women who drink.** In contrast to the ambivalent treatment bestowed on men who drink, the stigma around women was so strong that it led to acts of discrimination, concrete negative actions, violence, and poor treatment from members of their community:

> "*When a women drinks alcohol it is labeled as a sin, and this is followed by violence such as biting and harassment. Although men are treated as king, women are treated as slaves. It is obvious, to find gender violence in the community, especially when a women go against social norms*" (Interview #17, Male).

This treatment towards women stands in direct contrast to men who drank alcohol who were instead "*loved and appreciated by many*"; "*because drinking is seen as a prestige thing. . . [men were] applauded for drinking*" (Interview #16, Female).

Discrimination against women who drink arose mainly as isolation and came from fellow community members, loved ones, and family members alike:

> "*I can remember one lady who was well married and blessed with children, I don't know why and when she started drinking but the drinking brought so much trouble in her home to the extent she was divorced by her husband, and he took all her children away and went to marry*

*another woman. She really went through a rough time. People around were seeing her as insane as she can't take care of anything. She was not invited to join even in those women gatherings in the street or local church"* (Interview #10, Female).

*"I had one relative who married a woman who was initially not using alcohol, I don't know what happened; she started using alcohol so much and being aggressive. The family completely excluded her from any activity. . .from her husband, mother-in-law and even her children, she was not welcomed even to cook and eat together as a family"* (Interview #3, Female).

**Community support and implications on women's health.** Expressions of empathy or understanding towards women users were almost entirely absent across all interviews. Instead, women were said to be shown "*no support*" (Interview #1, Female) from family members or the community. As one female respondent said, *"the community does not even bother to find the reason as to why a woman drinks, they just go straight to judge that she has no manners"* (Interview #3, Female).

This gender-based stigma had direct implications on women's health outcomes. As one patient postulated, it prevented women from disclosing their alcohol use to their providers or other healthcare professionals – *"If a woman tells her healthcare provider that she has been drinking and feels judged, she might be less inclined to talk about this issue again"* (Interview #2, Male).

**Stigma's influence on sexual violence.** The disproportionate perceived and experienced stigma that women face from alcohol use was perceived to increase their risk of sexual assault and sexual violence. Because of how they were negatively viewed in society, women who consumed significant amounts of alcohol were seen as being unable, both physically and morally, to refuse men's sexual advances:

*"A woman who drinks too much alcohol, the community sees her as a drunkard. That woman who can be raped at any time when she is drunk. Any man can sleep with her while drunk and she cannot refuse to sleep with any man in need. But for the men who drink too much alcohol, the community feels it is okay. They don't look badly like a woman who drinks too much alcohol"* (Interview #6, Female).

## Alcohol and sexual violence

Although not included in the interview guide, when discussing the consequences of alcohol use, sexual assault was a common theme that arose across most interviews. In all cases except one, sexual assault was described as men inflicting sexual violence upon women, and alcohol was seen as a facilitator of this behavior.

**Women face disproportionate alcohol-related sexual harm.** Similar to stigma, women in Moshi were described as being disproportionately impacted by sexual violence, especially when under the influence of alcohol use. This sexual violence further put women in danger of contracting STDs, getting pregnant, or suffering additional physical and mental harm. Participants described that:

*"When a man gets drunk, there are some things which are not common like being raped, but. . .it's easy for a drunk woman to be a victim of such violence like being beaten, her things stolen, being raped, and forced to do some devastating things which at the end leads to unwanted pregnancies, or sexually transmitted infections like HIV and syphilis. That's how effects can be different between men and women"* (Interview #10, Female).

*"There was a day when I sent my son to buy groceries on his way back, he told me he saw a woman and a man taking off clothes near the local liquor club he said I don't know why, maybe the man wants to slaughter the woman"* (Interview #13, Female).

As is illustrated by these quotes, alcohol-facilitated sexual violence against women is present in the Moshi community and exists at significant harm to the women who drink. While the stigma associated with alcohol use was perceived to increase a woman's risk of experiencing sexual violence, she was also seen to be at greater risk of stigma and social harm after experiencing sexual assault. As described by one participant:

*"For a female, once they drink excessively, [they] may end up being raped or tortured by men. And once people know that something embarrassing has happened to you, everyone will point finger at you, also they will never respect you"* (Interview #14, Male).

**Alcohol is used as a tool to advance sexual acts.** Participants noted that some men utilize alcohol as a tool to intoxicate women to get women to sleep with them. While some described this as a method to decrease inhibitions, most respondents framed this as being able to "*easily take advantage of the girl*" (Interview #15, Female). Other respondents shared this view, stating:

*"Men I think are drinking alcohol to relax and others, through drinking alcohol, it is their way of getting women. When a man needs a woman, he goes to the bar. When he finds a woman alone, he starts buying alcohol for that woman in order to get her drunk and take her to sleep with him."* (Interview #9, Female)

*"Sometimes when a girl goes out with a man, I have seen men force girls to drink alcohol. A man will force a girl to drink at least just a glass of wine. Sometimes men do that to easily take advantage of the girl."* (Interview #15, Female).

Some women respondents also commented on their blunted decision-making capacities while under the influence of alcohol use. This led to sexual acts and other negative consequences that would normally be unwanted. As two described:

*"Women [who are] drunk may sleep with a man that she didn't want to if she was sober. This reckless decision may lead to things like acquiring diseases like sexually transmitted infections or unwanted pregnancies and can make this lady to go into depression as well"* (Interview #6, Female)

*"Many times, you will end up doing things you would not do in your right mind. Sometimes you may be irresponsible under the influence of alcohol, like sleeping with someone you don't know."* (Interview #15, Female)

**Women's alcohol use is associated with sex work.** Finally, women's alcohol use appeared to have a sociocultural link with sex work in some interviews. This relationship was noted even while direct interactions with sex workers were not present and could underlie the aforementioned connections between alcohol and sexual violence against women.

Two respondents elaborated on alcohol use in the context of sex work, saying those consumed alcohol to perform "*smoothly with confidence*" (Interview #3, Female).

**Table 2. Words used to describe unhealthy alcohol users.**

| | | Perceptions | |
|---|---|---|---|
| **Gender of Unhealthy Alcohol User** | | Men Respondents | Women Respondents |
| | Men Users | ❖ Cannot be involved in activities or any decision-making process in the community<br>❖ Irresponsible to cater family or social needs<br>❖ Incapable of delivering anything in the family<br>❖ Immature<br>❖ Hopeless<br>❖ They have no points<br>❖ Bullied by the community | ❖ Stigmatizing their families<br>❖ No real contribution in the family<br>❖ Don't respect themselves or their wives<br>❖ They have despised their families and gave alcohol the highest priority<br>❖ Burden to his family<br>❖ Silly drunkards who cannot make decisions<br>❖ Despised<br>❖ Lacks discipline<br>❖ No self-control<br>❖ Reckless<br>❖ Causes fights<br>❖ Society does not trust them with the works<br>❖ Not. . .constructive<br>❖ Segregated from making important decisions in the community<br>❖ Fellow men will not exclude him<br>❖ Strong<br>❖ Complete manhood |
| | Women Users | ❖ Irresponsible mothers<br>❖ Disgusting!<br>❖ She can't control herself<br>❖ Not well-mannered<br>❖ Disrespected<br>❖ Irresponsible<br>❖ They are after men<br>❖ She can be manipulated by other people, especially men<br>❖ Sex-workers | ❖ [Men] cannot marry a woman who drinks alcohol<br>❖ Not wife material<br>❖ Will never marry<br>❖ Spends a lot [of money]<br>❖ They will not take her seriously<br>❖ Ineffective<br>❖ Highly contemptible<br>❖ Not raised properly<br>❖ Careless people<br>❖ Not right at all<br>❖ Bad-mannered<br>❖ Not respectable<br>❖ Hopeless<br>❖ Irresponsible<br>❖ Disturbed<br>❖ People who like conflicts<br>❖ Bad spoken<br>❖ Despised<br>❖ Hated<br>❖ Hoe<br>❖ Prostitutes |

*"Speaking openly the job is prostitution, using her body to please someone and get paid in return. This brings money, but it's not an easy job to do, and in a sound mind it's not possible to do it. That's why they drink alcohol. Alcohol does excite the brain and makes you see everything is alright and after drinking a woman can do the job well, although she will regret later, at least she has gained the money for her needs already."* (Interview #10, Female)

In summary, distinct gender differences in the perceived acceptability of alcohol use arose consistently throughout in-depth interviews. An overview of these words and phrases, pulled from in-depth interviews is shown in Table 2. Words describing male alcohol users generally framed them as being irresponsible and unable to care for their family, but in some cases were complimentary, describing them as 'strong' and with 'complete manhood'. In comparison, words for women users were much harsher and unanimously negative. Participants painted women who drink as unfit to marry, affiliated them with sexual promiscuity, and notably, attacked their character and self-worth.

## Discussion

This is among the first studies to explore how alcohol-related stigma and sexual violence differ by gender among patients in Moshi, Tanzania. Our research extends previous work in Moshi that has identified high rates of local alcohol abuse [37, 39], gender differences in experienced stigma in injury patients [30], and an association between alcohol abuse and past experiences with sexual violence among women [36]. These findings add to the broader base of literature on alcohol intake in LMICs by sharing the critical role gender plays in societal expectations around its use. Through an in-depth qualitative analysis, we found that pervasive gender roles created distinct expectations for men's and women's daily responsibilities and alcohol intake, and underlined differences in alcohol-related stigma. The rigidity of gender roles here has important social implications that should be considered in future alcohol-related interventions and care. Primary among these, women in Moshi were found to face disproportionate social censure and isolation when drinking as compared to men. While this could deter intake, for women already suffering from unhealthy alcohol use, stigma hindered the honest reporting of use, discouraged treatment seeking, and removed the social support needed to abstain from further negative drinking habits. Men, in contrast, experienced little stigma and in fact some social benefit from drinking, a damaging norm that could facilitate heavy, problematic intake. Last, alcohol was seen to contribute to sexual violence, with men using alcohol to advance unwanted sexual acts that in almost all cases were to the detriment of women. As such, in addition to gender and stigma, future alcohol-related treatment programs should consider the association between alcohol use and sexual assault in designing alcohol-reduction interventions as this could make certain aspects of care more essential.

Our data present the rigidity of traditional gender roles within Moshi, with participants describing men as having greater monetary responsibilities while women were expected to oversee all household tasks. Our results are supported by existing research based in LMICs where traditional roles earmark men to work outside the home and women to work within it [16–18, 55]. For women, this social norm is known to limit their autonomy and financial decision-making power and has been linked with a greater prevalence of intimate partner violence and negative health outcomes [14, 17]. Further complicating this, our analysis revealed a belief that women should maintain constant, exemplary behavior and are expected to be infallible wives and mothers. However, quests for perfectionism and total infallibility have been linked with low self-esteem, low self-efficiency, anxiety, and depression when carried out in the context of critical environments [56, 57]. As explored more below, consequences for women were particularly severe when they were unable to conform to these expectations.

The presumption of perfection extended to alcohol consumption, where women, unlike men, were expected to go without alcohol entirely to properly care for their children and household. Globally, past literature and established international frameworks have suggested that variations in alcohol consumption behaviors may be partly attributable to traditional responsibilities for men and women [58, 59]. Research conducted by White et al., for example, suggested the gap between male and female alcohol use and alcohol-related harm has been narrowing in the United States alongside traditional gender roles [19, 60]. Interestingly, participants mentioned concern around men and women's drinking only as it would negatively affect their contribution to the labor force, but expressed no concern as to how it would affect their physical and mental health. This highlights an overlying societal view that the impact of alcohol use on labor was a prevailing concern among participants. The pronounced gender roles we identified, and the impact they likely wield on health outcomes given the existing evidence, means that research, health initiatives, and policy changes in Moshi should be especially cognizant of gender going forwards.

Another important finding was the disproportionate stigma women in Moshi faced from alcohol use even when intake occurred in lesser quantities than men. This finding has been similarly identified in the work by Griffin et al., who identified that perceived alcohol-related stigma was greater against female injury patients in Moshi [30], but also echoes larger gender inequities present in LMICs. As noted by Medine-Mora, women in LMICs are thought to experience greater social consequences from alcohol use as it typically does not align with the traditional behaviors expected of them [14, 61]. In our study, we found that stigma led to discrimination; women in Moshi with alcohol misuse garnered little understanding or sympathy from community members and instead were ostracized by friends and family. The stigma centered around women in Moshi is net negative as it prevents high-risk users from disclosing their alcohol use to healthcare providers, offset by a slight protective effect in dissuading alcohol intake for some. The effect of stigma on those with substance use disorders (SUDs) has been well studied globally, linking greater stigma with a lower likelihood of seeking treatment, lower quality of care from healthcare professionals, and overall poorer health outcomes [62, 63]. In Moshi, pervasive structural stigma may exacerbate these effects. Staton et al., for example, identified that 71% of healthcare practitioners in Moshi either discriminated against or devalued patients with alcohol use disorders, meaning that those patients likely received poorer care [34]. Further, as identified in this manuscript, the social ostracization of women who drink in Moshi negatively compounds their efforts to reduce alcohol use, as social support has been identified to be a vital factor in successful SUD recovery [64–66]. The stigma around women's alcohol use in Moshi places unique barriers to healthcare delivery and disease recovery for high-risk women users, which should be noted in ongoing research and programming efforts.

In contrast to the disproportionate stigmatization of women's drinking, men received little to no public censure for their alcohol use, even when obviously problematic. Public reproval was noted to fall on men's families, while the men themselves were still embraced and even sometimes admired for their drinking, a trivialization of problematic behavior that could facilitate excessive consumption. In Moshi, men are known to drink at higher rates than women and are more likely to be diagnosed with alcohol use disorder and experience alcohol-related injuries [38, 39, 42, 47]. Previous work by our team has likewise identified that in Moshi, social pressures contribute to men's drinking as their social interactions with other men typically occur in bars or other locations where drinking is encouraged [42]. Societal encouragement of men's alcohol consumption may be contributing to these high rates of use and negative impacts on men's health. As such, future efforts to minimize men's alcohol use should be considerate of social pressures to drink and incorporate provisions to address this important factor.

Along with the disproportionate burden of stigma, our data revealed that women are perceived to be at greater risk of experiencing sexual assault and violence while consuming alcohol than men. Respondents described alcohol use as facilitating men's infliction of sexual violence by intoxicating women. This led to reportedly little to no noted repercussions for men but severe consequences for women, including significant physical and emotional pain, stigma after assault, and unwanted pregnancy. Alcohol use and forced intoxication of the victim are known characteristics of perpetrators of sexual violence [67, 68]. In Sub-Saharan Africa, rates of HIV transmission and intimate partner violence (IPV) have been shown to increase with alcohol consumption [69–71]. Our results are also echoed in past research, where alcohol consumption was found to increase the risk of sexual assault for women and illuminates the growing concern that alcohol drives violent sexual behavior globally [70, 72–74].

The close relationship between alcohol use and sexual assault in our analysis shows the need for future interventions in Moshi to incorporate strategies aimed at reducing these major

issues. Programming to mitigate sexual violence and alcohol misuse should incorporate targeted solutions for both genders and include measures to raise awareness at a societal level. Interventions with a dual focus on reducing alcohol use and gender violence have already been piloted in several African countries, with generally positive results [75–78]. Murray et al., for example, implemented a psychotherapy-based treatment program in Zambia aimed at reducing hazardous alcohol use and IPV, which was found to be more effective than standard care alone in both aims [78]. While other interventions focus on education, empowerment, and motivational interviewing strategies, throughout all, as Wechsberg et al. highlight, it may be important to incorporate gender violence and other social determinants of health to optimize interventions [79].

As can be seen from this analysis, gender roles and disproportionate experiences of stigma and sexual violence are all vital factors that should be considered in designing and implementing effective and culturally appropriate alcohol-reduction programs in this region. Given the notable harm identified by our study participants, we recommend future researchers and policymakers work to develop interventions to fit the sociocultural context of Moshi and implement them accordingly. This advancement, in addition to addressing the lack of legal repercussions for perpetrators of sexual violence and raising societal awareness around stigma and alcohol use, will ensure greater support for and protection of the Moshi community. By incorporating this knowledge in future programming, the needs of alcohol users in this region can better be addressed, and alcohol-related harm can be more effectively minimized.

## Limitations

This analysis should be considered in light of several limitations. First is the lack of quantitative data with which the results presented here can be triangulated; while stigma and sexual assault are reported, we are unable to comment on the prevalence or quantify the risk of these outcomes in relation to alcohol use. However, our qualitative focus allows us to showcase patients' experiences and perspectives on these topics in a more focused and in-depth manner than would have been otherwise possible. Further, we believe that our choice to employ one-on-one, gender-matched in-depth interviews for data collection led to a more open and honest discussion on gender, stigma, and sexual assault, thus establishing more accurate results. Another limitation is that questions specifically pertaining to gender roles and sexual assault were not included in the interview guide, which means that some perspectives may not have been included from interviewees. This omission, however, also highlights the strength of our data collection process in allowing important topics to emerge organically and showcases how significant gender roles and sexual assault are in the context of local alcohol use. Given the impact gender roles in Moshi had on alcohol use and consequences, we recommend future research to explore the impact gender roles have on other health outcomes outside of alcohol misuse. Even when considering these limitations, this manuscript stands as the first, focused exploration of gender roles, alcohol-related stigma, and sexual assault through exclusively qualitative methods in Moshi, Tanzania. Our research provides a deeper understanding of how these consequences differentially affect men and women alcohol users in Moshi and showcase the need to consider these elements in future alcohol-reduction-focused programming.

## Conclusion

Men and women have differentiated social roles and expectations in Moshi, Tanzania, and, resultingly, have different experiences of the social consequences related to alcohol, including stigma and discrimination. Disproportionate stigma against women who drink facilitates

sexual violence while the trivialization of problematic drinking in men encourages excessive use. Future alcohol-related interventions and programming in this setting that consider the distinct gendered, social impacts of alcohol use would likely be more efficacious at lessening alcohol-related harm.

## Supporting information

**S1 Checklist. STROBE statement—Checklist of items that should be included in reports of observational studies.**
(DOCX)

**S1 Questionnaire. Inclusivity in global research.**
(DOCX)

## Author Contributions

**Conceptualization:** Alena Pauley, Judith Boshe, Catherine A. Staton, Blandina T. Mmbaga.

**Data curation:** Alena Pauley, Mariana Mikindo, Yvonne Sawe, Joseph Kilasara.

**Formal analysis:** Judith Boshe, Catherine A. Staton, Blandina T. Mmbaga.

**Funding acquisition:** Alena Pauley, Catherine A. Staton.

**Investigation:** Judith Boshe, Catherine A. Staton, Blandina T. Mmbaga.

**Methodology:** Alena Pauley, Catherine A. Staton.

**Supervision:** Sharla Rent, Catherine A. Staton, Blandina T. Mmbaga.

**Writing – original draft:** Alena Pauley, Madeline Metcalf, Mia Buono.

**Writing – review & editing:** Alena Pauley, Madeline Metcalf, Mia Buono, Sharla Rent, Catherine A. Staton.

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
