## [Decision Letter · Decision Letter 0]

14 Dec 2023

PGPH-D-23-01584

“When a man drinks alcohol it’s cool but when a woman drinks she is a hoe”: A Qualitative Exploration of Alcohol, Gender, Stigma, and Sexual Assault in Moshi, Tanzania

Dear Dr. Staton,

Thank you for submitting your manuscript to PLOS Global Public Health. After careful consideration, we feel that it has merit but does not fully meet PLOS Global Public Health’s publication criteria as it currently stands. Therefore, we invite you to submit a revised version of the manuscript that addresses the points raised during the review process.

We look forward to receiving your revised manuscript.

Kind regards,

Mrittika Barua

Academic Editor

Journal Requirements:

3. If you are reporting a retrospective study of medical records or archived samples, please ensure that you have discussed whether all data were fully anonymized before you accessed them and/or whether the IRB or ethics committee waived the requirement for informed consent. If patients provided informed written consent to have data from their medical records used in research, please include this information."

4. We have noticed that you have uploaded Supporting Information files, but you have not included a list of legends. Please add a full list of legends for your Supporting Information files after the references list.

Additional Editor Comments (if provided):

Reviewers' comments:

Reviewer's Responses to Questions

**Comments to the Author**

1. Does this manuscript meet PLOS Global Public Health’s publication criteria? Is the manuscript technically sound, and do the data support the conclusions? The manuscript must describe methodologically and ethically rigorous research with conclusions that are appropriately drawn based on the data presented.

Reviewer #1: Yes

Reviewer #2: Partly

2. Has the statistical analysis been performed appropriately and rigorously?

Reviewer #1: N/A

Reviewer #2: N/A

3. Have the authors made all data underlying the findings in their manuscript fully available (please refer to the Data Availability Statement at the start of the manuscript PDF file)?

Reviewer #1: Yes

Reviewer #2: Yes

4. Is the manuscript presented in an intelligible fashion and written in standard English?

Reviewer #1: Yes

Reviewer #2: Yes

5. Review Comments to the Author

Reviewer #1: An interesting and well written paper. This paper can give a new insight to understand context like alcolohol use from a gender lens, especially data collected through IDIs and analysis will help readers to understand the issue in depth. I would just suggest a minor revision that the IDI quotes need to be revised well. In some there are grammatical mistakes, which made meaning unclear.

Reviewer #2: Review

I would like to express my gratitude to the editor for giving me the opportunity to review this manuscript, which I very much enjoyed reading.

The authors present a study on a highly relevant, underresearched topic as part of a larger project of considerable importance. The title introduces the intricate interplay of alcohol use with dimensions on gender, stigma and violence, rendering the complexities of this massive global problem palpable through the voices of study participants.

The manuscript is very well written.

However, it has two major flaws that must be addressed, concerning its methodology and the discussion of findings.

Firstly, although the grounded theory approach is mentioned, there is a lack of appropriate reference and explanation. The description of methodology and findings raises doubts as to whether the authors have really applied grounded theory.

Secondly, the discussion is comprehensive and well written regarding the female angle, but it lacks depth as important findings are omitted. In particular, the absence of discussion on male behavior and societal perceptions is striking. A balanced examination of the implications, recommendations and conclusions for both genders and societal levels is warranted.

In addition to that, I have minor comments and suggestions for the authors’ consideration to enhance the quality of this work.

- Line 65, abstract: by reading the abstract I would expect a conclusion on both genders as the glorification of male drinking seems highly problematic and requires addressing

- Introduction: excellent, very clear and well written.

- Line 109: stigma as an established barrier – needs a reference

- Line 134: IDI needs to be introduced, however, I suggest spelling out IDI throughout for better readibility; the study already uses many abbreviations.

- Lines 154 ff: At this point, a section on reflexivity/positionality is needed in relation to the fact that the first author is not a local researcher and what led to this decision

- Line 178: point 3) – how can perspectives and experiences be known before data collection?

- Line 192: include info on transcription

- Line 206 ff: Please elaborate your step-wise approach with reference to key literature on grounded theory. There is insufficient description and, I’m afraid, probably also understanding of the grounded theory approach;

- Line 239: Consider using ‘findings’ or ‘analysis’ rather than ‘results’ for qual research

- Lines 246 ff: Nice summary of the findings, though not sure it is needed, consider removing if short on words

- Line 255, table: the table refers to themes and sub-themes rather than providing a grounded theory analysis, probably due to an overall misunderstanding of the stated research approach

- Line 554 ff: The discussion focuses mainly on the impact on women and not only ignores the other side of the findings: the focus on women also risks looking at the phenomenon in isolation, whereas I am convinced that male, female and societal behavior and attitudes are interconnected and have to be discussed and addressed comprehensively. The study and its findings are too important for these aspects to be left out.

I would like to see some discussion on, for example:

- Expectations are almost as if women had to behave like angels while being treated like inferior humans, slaves (sic); if their behaviour does not fully conform to societal norms, they are ousted

- Women’s drinking behaviour is only a concern because it affects their labour force, society doesn’t seem to care about how alcohol affects women’s health and well-being

- the same is true for men, except that in their case heavy drinking is either ignored or glorified, a hugely damaging trivialization at the detriment of men’s health

- thorough elaboration on the problematic behaviour of men, both in terms of their alcohol consumption and their use of alcohol to justify and perpetrate SGBV against alcoholised women

- eg. line 618: alcohol use facilitate this behaviour, but also the lack of sanctions/punishment => programmes must focus on support of affected women plus legal consequences for men plus changing societal attitudes

- Programmes to mitigate this problem need to provide solutions for both genders and include awareness raising measures at a societal level

- Line 649: can also be seen as a strength that – although not explicitly asked - the data collection process allowed this important topic to emerge

- Lines 660 ff: adjust conclusion taking into account the above points

6. PLOS authors have the option to publish the peer review history of their article (what does this mean?). If published, this will include your full peer review and any attached files.

**Do you want your identity to be public for this peer review?** For information about this choice, including consent withdrawal, please see our Privacy Policy.

Reviewer #1: **Yes: **Sanzida Akhter

Reviewer #2: No

---

## [Editor Report · Decision Letter 1]

6 Feb 2024

“When a man drinks alcohol it’s cool but when a woman drinks she is a hoe”: A Qualitative Exploration of Alcohol, Gender, Stigma, and Sexual Assault in Moshi, Tanzania

PGPH-D-23-01584R1

Dear Dr. Staton,

We are pleased to inform you that your manuscript '“When a man drinks alcohol it’s cool but when a woman drinks she is a hoe”: A Qualitative Exploration of Alcohol, Gender, Stigma, and Sexual Assault in Moshi, Tanzania' has been provisionally accepted for publication in PLOS Global Public Health.

Best regards,

Mrittika Barua

Academic Editor